# RTC-VAE: Harnessing the peculiarity of Total Correlation in Learning Disentangled Representations

## Abstract

In the problem of unsupervised learning of disentangled representations, one of the promising methods is to penalize the total correlation of sampled latent variables. Unfortunately, this well-motivated strategy often fails to achieve disentanglement due to a problematic difference between the sampled latent representation and its corresponding mean representation. We provide a theoretical explanation that low total correlation of sample distribution cannot guarantee low total correlation of the mean representation. We prove that for the mean representation of arbitrarily high total correlation, there exist distributions of latent variables of a bounded total correlation. However, we still believe that total correlation could be a key to the disentanglement of unsupervised representative learning, and we propose a remedy, RTC-VAE, which rectifies the total correlation penalty. Experiments show that our model has a more reasonable distribution of the mean representation compared with baseline models, e.g., $\beta$-TCVAE and FactorVAE.

## 1 Introduction

VAEs (Variational AutoEncoders) Kingma & Welling (2013); Bengio et al. (2007) follow the common assumption that the high-dimensional real world observations $\mathbf{x}$ can be re-generated by a lower-dimension latent variable $\mathbf{z}$ which is semantically meaningful. Recent works Kim & Mnih (2018); Chen et al. (2018); Kumar et al. (2017) suggest that decomposing the ELBO (Evidence Lower Bound) could lead to distinguishing the factor of disentanglement. In particular, recent works Kim & Mnih (2018); Chen et al. (2018) focused on a term called total correlation (TC). The popular belief Chen et al. (2018) is that by adding weights to this term in objective function, a VAE model can learn a disentangled representation. This approach appears to be promising since the total correlation of a sampled representation should describe the level of factorising since total correlation is defined to be the KL-divergence between the joint distribution $\mathbf{z} \sim q(z)$ and the product of marginal distributions $\prod_j q(z_j)$. In this case, a low value suggests a less entangled joint distribution. However, Locatello et al. (2018) pointed out that the total correlation of sampled distribution $TC_{sample}$ being low does not necessarily give rise to a low total correlation of the corresponding mean representation $TC_{mean}$. Conventionally, the mean representation is used as the encoded latent variables, an unnoticed high $TC_{mean}$ is usually the culprit behind the undesirable entanglement. Moreover, Locatello et al. (2018) found that as regularization strength increases, the total correlation of sampled representation $TC_{sample}$ and mean representation $TC_{mean}$ are actually negatively correlated.

Locatello et al. (2018) put doubts on most methods of disentanglement including penalizing the total correlation term Kim & Mnih (2018); Chen et al. (2018), and they concluded that "the unsupervised learning of disentangled representations is fundamentally impossible without inductive biases".

Acknowledging the difficulty in learning disentangled representation, we provide a detailed explanation of the seemingly contradictory behaviors of the total correlations of sampled and mean representation in previous works on TC penalizing strategy. Moreover, we find that this problem described above can be remedied simply with an additional penalty term on the variance of a sampled representation.

Our contributions:

- In Theorem 1, we prove that for all mean representations, there exists a large class of sample distributions with bounded total correlation. Particularly, a mean representation with arbitrarily large total correlation can have a corresponding sample distribution with low total correlation. This implies that a low total correlation of sample distribution cannot guarantee a low total correlation of the mean representation. (Section. 2)

- Acknowledging the issue above, we further delve into total correlation, and provide a simple remedy by adding an additional penalty term on the variance of sample distribution. The penalty term forces a sampled representation to behave similar to the corresponding mean representation. Such penalty term is necessary for the strategy of penalizing $TC_{mean}$ in the view of Theorem 1. (Section. 4)

- We study several different methods of estimating total correlation. They are compared and benchmarked against the ground truth value on the multivariate Gaussian distribution Locatello et al. (2018). We point out that the method of (minibatch) estimators suffers from the curse of dimensionality and other drawbacks, making their estimation accuracy decay significantly with the increase of the dimension of the latent space, and some strong correlated distributions can be falsely estimated to have low total correlation. (Section. 5)

## 2 THE PECULIARITY OF TOTAL CORRELATION

In information theory, total correlation is one of the generalizations of mutual information, which measures the difference between the joint distribution of multiple random variables and the product of their marginal distributions. A high value means the joint distribution is far from an independent distribution, and hence it suggests high entanglement among these random variables.

**Definition 1.** *Total correlation of random variable* $\mathbf{x}$,

$$\mathrm{TC}(\mathbf{x}) := \mathrm{KL}\left(p(x)||\prod_j p(x_j)\right) = \mathbb{E}_{p(x)}\left[\log \frac{p(x)}{\prod_j p(x_j)}\right].$$

Naturally, people seek the solution of disentanglement in the form of low total correlation of the latent variables, e.g. Kim & Mnih (2018); Chen et al. (2018). However, there can be large difference between the total correlations of sample representation and mean representation. Forcing the former to be small does not guarantee the latter being small. In fact, given a mean representation of arbitrarily large total correlation, we can construct a family of distribution of sample representation that have a bounded total correlation, where the bound does not rely on the total correlation of the mean.

**Theorem 1.** *Let* $\boldsymbol{\mu} \sim \mathcal{N}(0, \boldsymbol{\Sigma})$ *and* $\sigma_j$ *be the standard deviation of* $\mu_j$, $j = 1, \cdots, D$, *and* $\max_j \sigma_j = c_0$. *For a fixed* $\mu$, *let* $\mathbf{z} \sim \mathcal{N}(\mu, \boldsymbol{\Sigma}'(\mu))$, *where* $\boldsymbol{\Sigma}'(\mu)$ *is diagonal and satisfies that for some* $R > 0$,

$$\begin{cases} c_2 > \sigma_j'(\mu) > c_1 > 0, & \text{if } |\mu| < R, \\ c_3 > \sigma_j'(\mu) > \dfrac{c_4}{|\mu|^l}, \text{ for some } l \geq 1, & \text{if } |\mu| > R. \end{cases} \tag{1}$$

*for some constants* $c_1, c_2, c_3, c_4$. *Then* $\mathrm{TC}(\mathbf{z}) \leq C$ *for some* $C = C(R, c_0, \cdots, c_4, l) > 0$.

The details of the proof are presented in Appendix A.3. Here's another way to interpret Theorem 1: with $C$ and parameters $R, c_0, \cdots, c_4, l$ fixed, one can make $TC(\boldsymbol{\mu})$ arbitrarily large, since $TC(\boldsymbol{\mu})$ depends only on the correlation matrix of $\boldsymbol{\mu}$ (see Proposition 1).

Theorem 1 provides an explanation to the contradiction observed by Locatello et al. (2018) that $\mathrm{TC}(\mathbf{z})$ is low does not mean $\mathrm{TC}(\boldsymbol{\mu})$ is low (actually much higher than $\mathrm{TC}(\mathbf{z})$). Since there exist such a large class of distributions of $\mathbf{z}$ that all have bounded $TC(\mathbf{z})$. When the objective function only penalizes $TC(\boldsymbol{\mu})$, neural networks are so flexible to easily find a distribution with low $TC(\mathbf{z})$, and total correlation estimators like MSS can encourage shutting down latent dimensions (see Section 5.3), which together cause the disparity of $TC(\boldsymbol{\mu})$ and $TC(\mathbf{z})$. This fact was not noticed until Locatello et al. (2018), and our investigation gives an explanation of the peculiar property of total correlation. Hence, Theorem 1 leads to the necessity of a regularizer of the difference between the

distributions of $\boldsymbol{\mu}$ and $\mathbf{z}$ when penalizing $TC_{sample}$. In Section. 4, we propose a simple regularizer that serves this goal.

It is an interesting question whether there exists a distribution of $\mathbf{z}$ $\mathcal{N}(\mu, \boldsymbol{\Sigma}'(\boldsymbol{\mu}))$ with arbitrarily small $TC(\mathbf{z})$ given $\mu$. If not, what's the lower bound of $TC(\mathbf{z})$? These questions remain open to us for now, and we leave them to future work.

## 3    RELATED WORKS

In the study of disentanglement, Higgins et al. (2017) proposed a modification of the VAE framework and introduced an adjustable hyperparameter $\beta$ that balances latent channel capacity and independence constraints with reconstruction accuracy. One drawback of $\beta$-VAE is the trade-off between the reconstruction quality and disentanglement. Motivated to alleviate this trade-off of $\beta$-VAE, Kim & Mnih (2018) proposed FactorVAE which decomposes the evidence lower bound and penalize a term measuring the total correlation between latent variables. Around the same time, Chen et al. (2018) proposed a similar ELBO decomposition method called $\beta$-TCVAE. The major difference between FactorVAE and $\beta$-TCVAE lies in their different strategies of estimating total correlation. Chen et al. (2018) used formulated estimators while Kim & Mnih (2018) utilized the density-ratio trick which requires an auxiliary discriminator network and an inner optimization loop. We will discuss these two strategies more in details in Section. 5.

The works above belong to representative learning without inductive biases. There are also works about representative learning with inductive biases, see Rolinek et al. (2019) and references therein.

As for the disentanglement metric, this will be discussed in Section. 6.

## 4    RECTIFIED-TCVAE

To simplify notation, let $p(n) = p(x_n)$, $q(z|x_n) = q(z|n)$. Recall the average *evidence lower bound* (ELBO),

$$\text{ELBO} := \mathbb{E}_{p(n)}\left[\mathbb{E}_{q(z|n)}[\log p(n|z)] - \text{KL}(q(z|n)\|p(z))\right]. \tag{2}$$

Chen et al. (2018) and independently by Kim & Mnih (2018) introduced objective function that penalizes total correlation, which can be formulated as

$$\mathcal{L}_{\beta-\text{TC}} := \text{ELBO} - \beta\text{TC}(z). \tag{3}$$

This approach unfortunately has a drawback. It turns out that instead of being able to obtain disentangled representation, we often find a sample representation appears to be disentangled while the mean representation is still entangled. In fact, when we are maximizing $\mathcal{L}_{\beta-\text{TC}}$, we could totally end up learning a distribution of $\mathbf{z}$ that makes $\text{TC}(\mathbf{z})$ goes low, while the total correlation of its mean $\boldsymbol{\mu}$ is still high. To resolve this, we define RTC-VAE,

$$\mathcal{L}_{\text{RTC}} := \mathcal{L}_{\beta-\text{TC}} - \eta \cdot \text{tr}(\mathbb{E}_{p(n)}Cov_{q(z|n)}[z]), \tag{4}$$

where

$$\text{tr}(\mathbb{E}_{p(n)}Cov_{q(z|n)}[z]) = \sum_{k}^{D}\mathbb{E}_{p(n)}[\sigma_k^2(n)].$$

Our penalty originates from the first term of the law of total covariance $Cov_{q(z)}[z] = \mathbb{E}_{p(n)}Cov_{q(z|n)}[z] + Cov_{p(n)}(\mathbb{E}_{q(z|n)}[z])$. Factorized representation[1] indicates a diagonal covariance matrix $Cov_{q(z)}[z]$. Motivated by this, Kumar et al. (2017) penalizes the off-diagonal terms in the second term, while ignores $\mathbb{E}_{p(n)}Cov_{q(z|n)}[z]$ since it is diagonal. Their penalty term leads to a vanishing $\mu$, which is the mean representation. The remedy to this is to add another penalty term on the distance between $\mu$'s and 1. DIP-VAE Kumar et al. (2017) employs this remedy, however, DIP-VAE does not outperform other VAE's when measured by various disentanglement metrics, e.g., FactorVAE score, see Fig. 3 & 14 in Locatello et al. (2018). This is actually not surprising since the

---

[1]Factorized representation means each latent dimension is independent.

two penalty terms in DIP-VAE contribute in opposite directions, with one leading to vanishing $\mu$'s and another fighting against it. This formulation can easily get the model stuck in saddle points.

Our objective, on the other hand, does not penalize directly on $\mu$. Instead, it penalizes on $\sigma$, the standard deviation of the distribution $q(z|n)$. This may seem little counter-intuitive at first sight, since penalizing a diagonal component of covariance $Cov[\mathbf{z}] = Cov_{q(z)}[z]$ seems not helpful to factorising. However, in the view of Theorem 1, our objective will force the distribution of $\mathbf{z}$ to be similar to the distribution of $\boldsymbol{\mu}$. Hence, it pushes us away from the situation of large $\text{TC}(\boldsymbol{\mu})$ and low $\text{TC}(\mathbf{z})$. Consequently, by minimizing $\text{TC}(\mathbf{z})$ we get a model that has low $\text{TC}(\boldsymbol{\mu})$, a disentangled mean representation.

## 5 ESTIMATION OF TOTAL CORRELATION

The naive Monte Carlo method comes with an intrinsic issue of underestimating total correlation. To avoid or resolve this, Kim & Mnih (2018) proposed a discriminator network with the help of *density-ratio trick* (see equation (3) and Appendix D. of Kim & Mnih (2018)). In Chen et al. (2018), two kinds of estimator of total correlation are proposed, Minibatch Weighted Sampling (MWS) and Minibatch Stratified Sampling (MSS) (see Appendix C.1 and C.2 in Chen et al. (2018)).

### 5.1 METHOD OF MINIBATCH ESTIMATORS

For instance, MSS can be described as followed. For a minibatch of sample, $B_{M+1} = \{n_1, \ldots, n_{M+1}\}$,

$$\mathbb{E}_{q(z,n)}[\log q(z)] \approx \frac{1}{M+1} \sum_{i=1}^{M+1} \log f(z_i, n_i, B_{M+1} \setminus \{n_i\}), \tag{5}$$

where[2]

$$f(z, n^*, B_{M+1} \setminus \{n^*\}) = \frac{1}{N} q(z|n^*) + \frac{1}{M} \sum_{m=1}^{M-1} q(z|n_m) + \frac{N-M}{NM} q(z|n_m). \tag{6}$$

For the convenience of readers, MWS is listed in Appendix A.1.

### 5.2 METHOD OF DENSITY-RATIO TRICK AND DISCRIMINATOR

Density-ratio trick Nguyen et al. (2010); Sugiyama et al. (2012) can be used to estimate KL-divergence,

$$\text{TC}(\mathbf{z}) = \text{KL}(q(z) \| \prod_j q(z_j)) = \mathbb{E}_{q(z)} \left[ \log \frac{q(z)}{\prod_j q(z_j)} \right] \tag{7}$$

$$\approx \mathbb{E}_{q(z)} \left[ \frac{D(z)}{1 - D(z)} \right], \tag{8}$$

where $D$ is discriminator that classifies $\mathbf{z}$ being sampled from $q(z)$ or $\prod_j q(z_j)$. For detail implementation, please refer to section 3 in Kim & Mnih (2018).

### 5.3 COMPARISON OF THE TWO METHODS

For multivariate normal distribution, the total correlation can be explicitly calculated which can be used as a ground truth for our comparison. To be specific,

**Proposition 1.** *Let* $\mathbf{x} \sim \mathcal{N}(0, \boldsymbol{\Sigma})$*, then*

$$\text{TC}(\mathbf{x}) = \frac{1}{2} \left( \log|\text{diag}(\boldsymbol{\Sigma})| - \log|\boldsymbol{\Sigma}| \right). \tag{9}$$

---

[2]There is a small part of the implementation of MSS in Chen et al.'s code that is not quite clear to us, specifically, the computation of log importance weight matrix in equation 6. In our experiment, we implement MSS with our understanding and denote it as $\text{MSS}_1$, and we denote Chen et al.'s implementation $\text{MSS}_0$. See Appendix A.2

It's difficult to track the exact reference of Proposition 1 since it is a fundamental property in information theory. Locatello et al. (2018) used this proposition to approximate the total correlation of the mean representation in latent space. In appendix, we provide a simple proof for the convenience of the readers.

We compared the performance of each method, MWS, $MSS_0$ and $MSS_1$ on the estimation of total correlation. For $\boldsymbol{\mu} \sim \mathcal{N}(0, \mathbf{I})$, and $\mathbf{z}|\mu \sim \mathcal{N}(\mu, \boldsymbol{\Sigma})$ where $\boldsymbol{\Sigma} = \mathrm{diag}(\sigma^2)$ and $\sigma = 0.1$. We choose $\sigma$ small so that the distribution of $\mathbf{z}$ can be approximated by normal distribution. Results are presented in Figure 1.

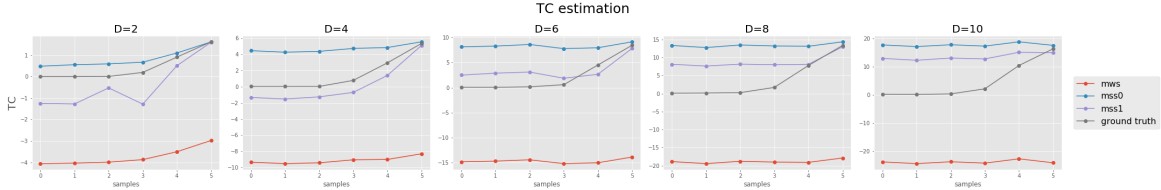

Figure 1: Different estimators of total correlation vs. ground truth on latent space of dimension low to high.

From Figure. 1, we can summarize the following observations: 1. MWS tends to underestimate total correlation in general; 2. For latent space of dimension $\leq 4$, $MSS_0$ and $MSS_1$ are quite accurate; 3. For latent space of high dimension, both $MSS_0$ and $MSS_1$ tend to overestimate total correlation when the actual value of total correlation is small; 4. Overall $MSS_1$ estimates closer to ground truth than $MMS_0$ does.

In the following analysis of the above observations, we use a less formal way of analyzing, which can be formalized to be rigorous, in order to convey our idea directly.

To interpret the third observation, let $\boldsymbol{\mu} \sim \mathcal{N}(0, \mathrm{Id})$ and $\mathbf{z}|\mu \sim \mathcal{N}(0, \boldsymbol{\Sigma})$, where Id is identity matrix and $\boldsymbol{\Sigma} = \mathrm{diag}(\sigma^2)$. Then $TC(\boldsymbol{\mu}) = 0$ and $TC(\mathbf{z})$ small if $\sigma$ small. Consider $q(z_k^{(i)}|n^{(j)})$ where $(i, j, k)$ are indices of a cube $(minibatch, minibatch, dimension)$ with size $M \times M \times D$ and $n^{(j)}$ is a sample drawn in a minibatch and $z^{(i)} = z(n^{(i)})$. We claim this: when the ground truth total correlation of $\mathbf{z}$ is low (the off-diagnal values of correlation matrix is small), only the elements on the diagonal surface of the cube, namely those elements of index $(i, i, k)$, take some bounded values $O(1)$, and all the other elements are very small $o(1)$ (since $\sigma = 0.1$).

To see the claim, let's first consider 1-D case, where $\boldsymbol{\mu} \sim \mathcal{N}(0, 1)$, $\mathbf{z}|\mu \sim \mathcal{N}(0, \sigma^2)$. When $\sigma$ is small, $\mathbf{z}$ can be approximately treated as $N(0, 1)$. $z^{(i)}$ and $\mu^{(j)}$ are independent for $i \neq j$, hence $z^{(i)} - \mu^{(j)} \sim \mathcal{N}(0, 2)$, and

$$P(|z^{(i)} - \mu^{(j)}| < t) = \frac{t}{2} + O(t^2) \tag{10}$$

See a proof in Appendix A.5. Then for D-dimension, the probability $P(|z^{(i)} - \mu^{(j)}| < t)$ would be $O(t^D)$. Now, only if $t$ takes value as small as $\sigma$, $q(z_k^{(i)}|n^{(j)})$ is not small. For example, $\sigma = 0.1$ and $D = 10$ and the chance of such case to happen is $O(10^{-10})$. Compared to batchsize, usually $O(10^3)$, the amount of such cases can be ignored.

Hence,

$$
\begin{aligned}
TC(\mathbf{z}) &= \mathbb{E}_{q(z)}\left[\log \frac{q(z)}{\prod_k q(z_k)}\right] \\
&= \mathbb{E}_{q(z,n)}[\log q(z)] - \mathbb{E}_{q(z,n)}[\log \prod_k q(z_k)] \\
&\approx \frac{1}{M}\sum_i \left(\log \frac{1}{M}\sum_j \prod_k q(z_k^{(i)}|n^{(j)}) - \log \prod_k \frac{1}{M}\sum_j q(z_k^{(i)}|n^{(j)})\right) \\
&\approx \frac{1}{M}\sum_i \left(\log \frac{1}{M}\sum_{j=i} O(1) - \log \prod_k \frac{1}{M}\sum_{j=i} O(1)\right) \\
&\approx \frac{1}{M}\sum_i \left(\log O(\frac{1}{M}) - \log O(\frac{1}{M^D})\right) \\
&\approx O((D-1)\log M).
\end{aligned}
$$

Assigning weights to elements $q(z_k^{(i)}|n^{(j)})$ such as MSS does not make essential change to the analysis above.

In addition, $\beta$-TCVAE (trained with MSS in our experiments and with MWS in Locatello et al. (2018)) seems to have an increasing total correlation of mean representation as regularization strength increases (higher $\beta$'s), as observed by Locatello et al. (2018). Here, we provide an explanation to the cause of this problem:

First, MSS and MWS prefer to shut down latent dimensions, meaning that distributions with fewer active dimensions can score lower estimated total correlation. Consider that $\boldsymbol{\mu}_0 \sim \mathcal{N}(0, 0.01)$, $\boldsymbol{\mu}_{0-} \sim \mathcal{N}(0, \mathrm{Id})$ and also $\mathbf{z}|\mu \sim \mathcal{N}(0, \boldsymbol{\Sigma})$, where $\boldsymbol{\Sigma} = \mathrm{diag}(\sigma^2)$. Then all elements of index $(i, j, 0)$, i.e., $q(z_0^{(i)}|n^{(j)})$ are not small, say $O(1)$, for the same reason as previous in $D-1$ dimension latent space. Thus, $\frac{1}{M}\sum_j q(z_0^{(i)}|n^{(j)}) \approx \frac{1}{M}\sum_j O(1) \approx O(1)$, and

$$
\begin{aligned}
TC(\mathbf{z}) &= \mathbb{E}_{q(z)}\left[\log \frac{q(z)}{\prod_k q(z_k)}\right] \\
&\approx \frac{1}{M}\sum_i \log \frac{\frac{1}{M}\sum_j \prod_k q(z_k^{(i)}|n^{(j)})}{\prod_k \frac{1}{M}\sum_j q(z_k^{(i)}|n^{(j)})} \\
&\approx \frac{1}{M}\sum_i \log \frac{\frac{1}{M}\sum_j (q(z_0^{(i)}|n^{(j)}) \cdot \prod_{k>0} q(z_k^{(i)}|n^{(j)}))}{\prod_{k>0} \frac{1}{M}\sum_j q(z_k^{(i)}|n^{(j)})} \\
&\approx \frac{1}{M}\sum_i \log \frac{\frac{1}{M}\cdot O(1)}{\prod_{k>0}\frac{1}{M}\cdot O(1)} \\
&\approx \log O(M^{D-2}) \\
&\approx O((D-2)\log M),
\end{aligned}
$$

compared to $O((D-1)\log M)$ when $\boldsymbol{\mu} \sim \mathcal{N}(0, \mathrm{Id})$ in previous analysis.

Now, consider any strongly correlated $\mathbf{z}$'s (e.g., $(\mathbf{z}_1, \mathbf{z}_2) \sim \mathcal{N}(0, \boldsymbol{\Sigma})$, where $\boldsymbol{\Sigma} = \begin{pmatrix} 0.01 & -0.1 \\ -0.1 & 1 \end{pmatrix}$, see Figure 2). Then the Gaussian (ground truth) total correlation is arbitrarily large ($\mathrm{TC}(\mathbf{z}_1, \mathbf{z}_2) = \infty$). This kind of distribution can score a relatively low TC value (for instance lower than $\mathbf{z}$) with estimators such as MSS and MWS by the analysis above. Hence, as $\beta$ increases, VAE trained with these estimators will be encouraged to obtain some dimensions of very low variance, and these dimensions are easily trapped in a strong correlation with other dimensions (like $\mathbf{z}_1$ and $\mathbf{z}_2$).

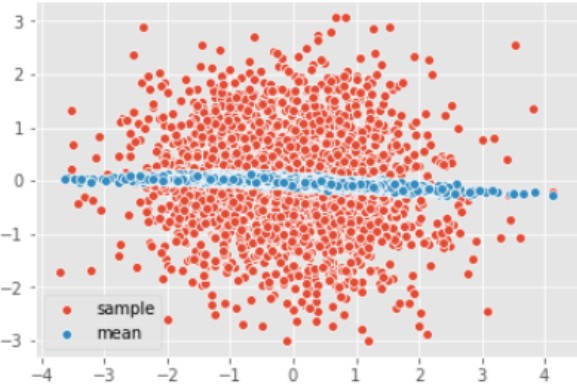

Figure 2: One dimension of mean has low variance (shutting down), and the distribution is strongly correlated (It appears to be almost flat due to small scale of the shutdown dimension). A sampled distribution (e.g. $\mathbf{z}|\mu \sim \mathcal{N}(\mu, 1)$) has a very low TC.

Shutting down dimension is not preferable because latent dimensions should not be fewer than ground truth. Moreover, considering datasets such as dSprites, though shape is labelled as a single dimension, models can learn to represent complex geometry with multiple dimensions, hence more active dimensions are learned than the number of ground-truth dimensions.

Based on the reasons above, we opt for the method of discriminators (density-ratio trick) in our implementation. Experiment shows that density-ratio trick provides a more stable estimation of total correlation when training VAE.

## 6 EXPERIMENTS

The datasets we use include dSprites Matthey et al. (2017), Shapes3D Burgess & Kim (2018) and 3D faces Paysan et al. (2009). At the time of writing, the scale of our experiments is limited, but there are already some evidence to deliver our arguments. We have scheduled further experiments and tests on larger scale in future works.

For every model, we trained with 10 different initialization. Hyperparameter $\beta$, also the regularization strength, takes $2, 4, \cdots, 10$. For hyperparameter $\eta$, we fix $\eta = 10$ for a simple reason. Since the variance term in equation 4 becomes small (close to 0) shortly after training begins, this term will not contribute much compared with $\mathcal{L}_{beta-TC}$ term. So, $\eta = 10$ is enough to strengthen the penalty at the beginning of training. In our experiments, higher value of $\eta$ cannot bring further improvement since $\mathbf{z}$ is close to $\boldsymbol{\mu}$ already. And lower values may not guarantee $\mathbf{z}$ being close to $\boldsymbol{\mu}$.

From experiments, we observe that RTC-VAE has much lower $TC_{mean}$ with different regularization strength than FactorVAE does (Figure 4). And on different datasets, this is also the case (see Appendix). The $TC_{mean}$ behaves almost identically as $TC_{sample}$ in RTC-VAE (see Figure 3 (b)). The problem of contradictory behaviors of $TC_{mean}$ and $TC_{sample}$ is evidently remedied by RTC-VAE. In addition, the ELBO of RTC-VAE seems to converge faster than FactorVAE as a byproduct (see Figure 5 and Figure 6). Examining the distributions of latent dimensions (mean representation), FactorVAE tends to have some strongly correlated latent dimensions (see Figure 8), and RTC-VAE shows well factorized latent distributions (see Figure 7).

### 6.1 METRICS OF DISENTANGLEMENT

We wish there were a widely accepted metric of disentanglement to compare our model RTCVAE with other models. Unfortunately, it is still an open question, how we can measure disentanglement. Various attempts have been made so far, but Locatello et al. (2018) challenged most of them, indicating that the score under any metric varies due to different initializations and datasets. Here,

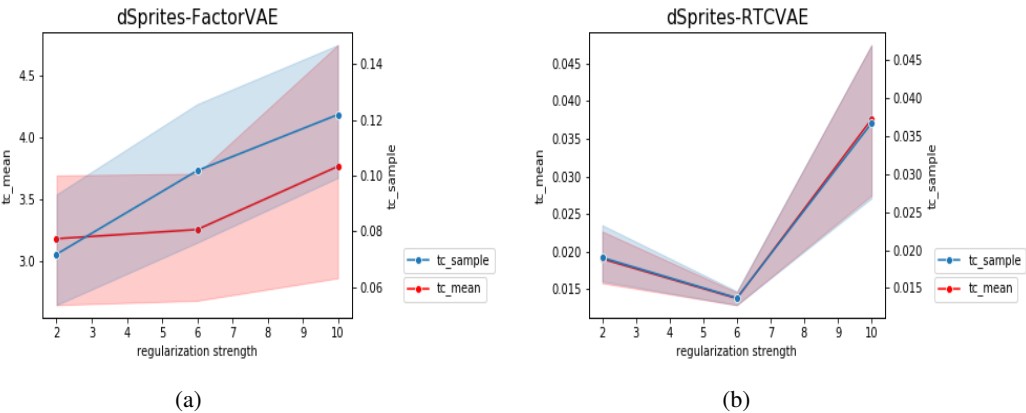

Figure 3: The shaded region indicates $90\%$ confidence interval. (a) The total correlation of sample and mean representation of FactorVAE on dSprite with $\beta = 2, 6, 10$. There is a large difference in scales of $TC_{sample}$ and $TC_{mean}$. (b) The total correlation of sample and mean representation of RTCVAE on dSprite with $\beta = 2, 6, 10$ and $\eta = 10$. There is almost no difference between $TC_{sample}$ and $TC_{mean}$ due to the variance penalty term in equation 4.

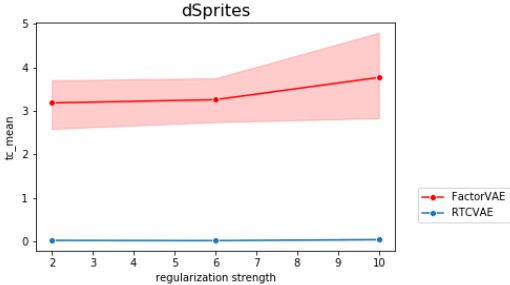

Figure 4: Direct comparison between the $TC_{mean}$ of FactorVAE and RTCVAE.

we analyse several important metrics and attempt to point out some blind spots that have not being considered by these metrics.

Higgins et al. (2017) proposed using a classifier to measure each dimension of latent space and each ground truth factor, e.g. $(x, y)$ coordinates, scale, rotation, etc. Kim & Mnih (2018) revised this

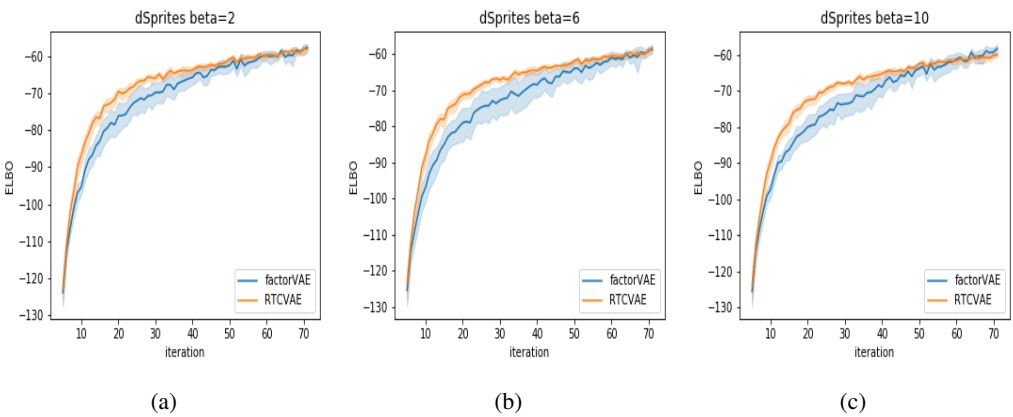

Figure 5: Training ELBO of RTCVAE and FactorVAE.

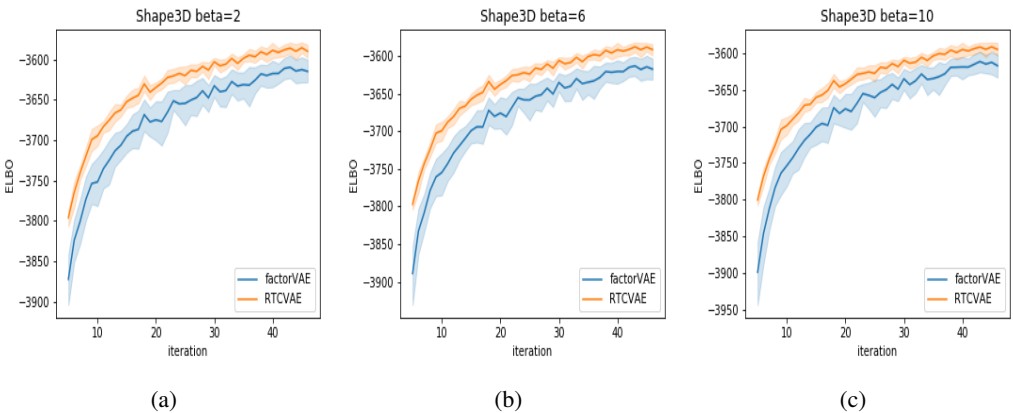

Figure 6: Training ELBO of RTCVAE and FactorVAE on Shapes3D.

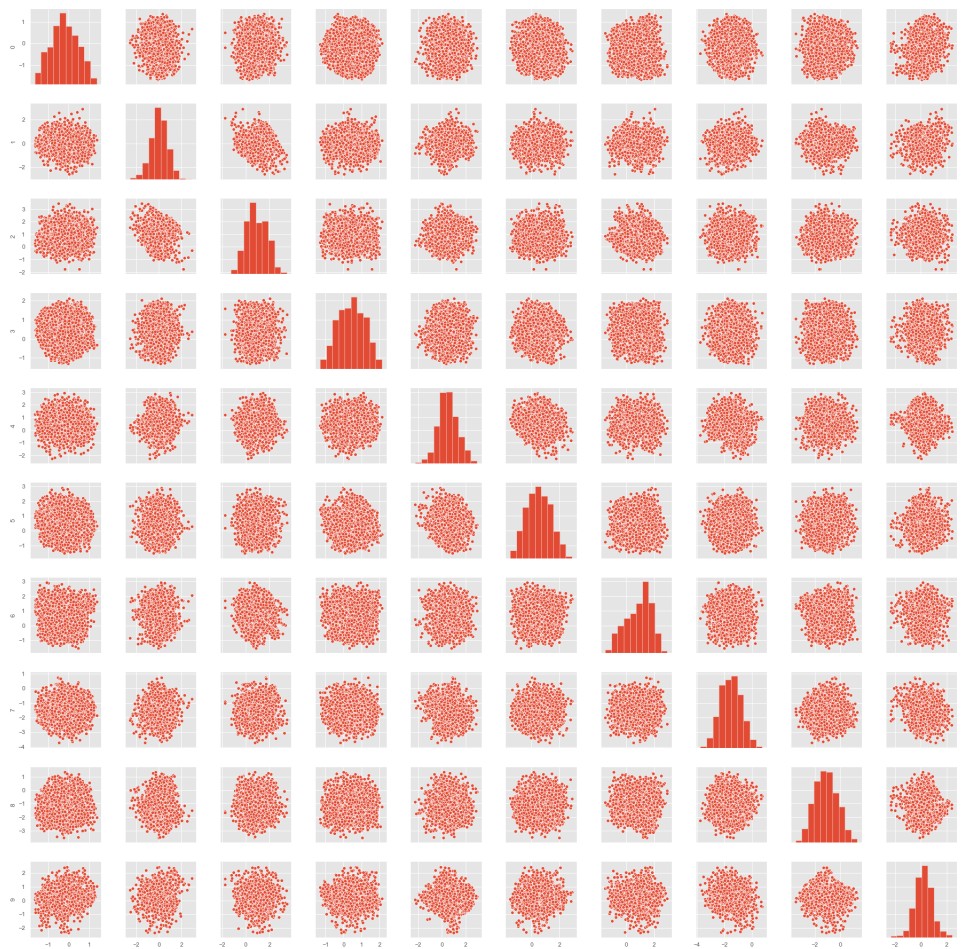

Figure 7: The pairplot of 10 latent dimensions of RTCVAE on Shapes3D. No dimensions show strong correlation.

approach yet kept the idea. Chen et al. (2018) introduced mutual information gap (MIG), which estimates the mutual information between each latent dimension and each ground truth factor. Note that this is a classifier-free metric. Eastwood & Williams (2018) proposed a framework of disentan-

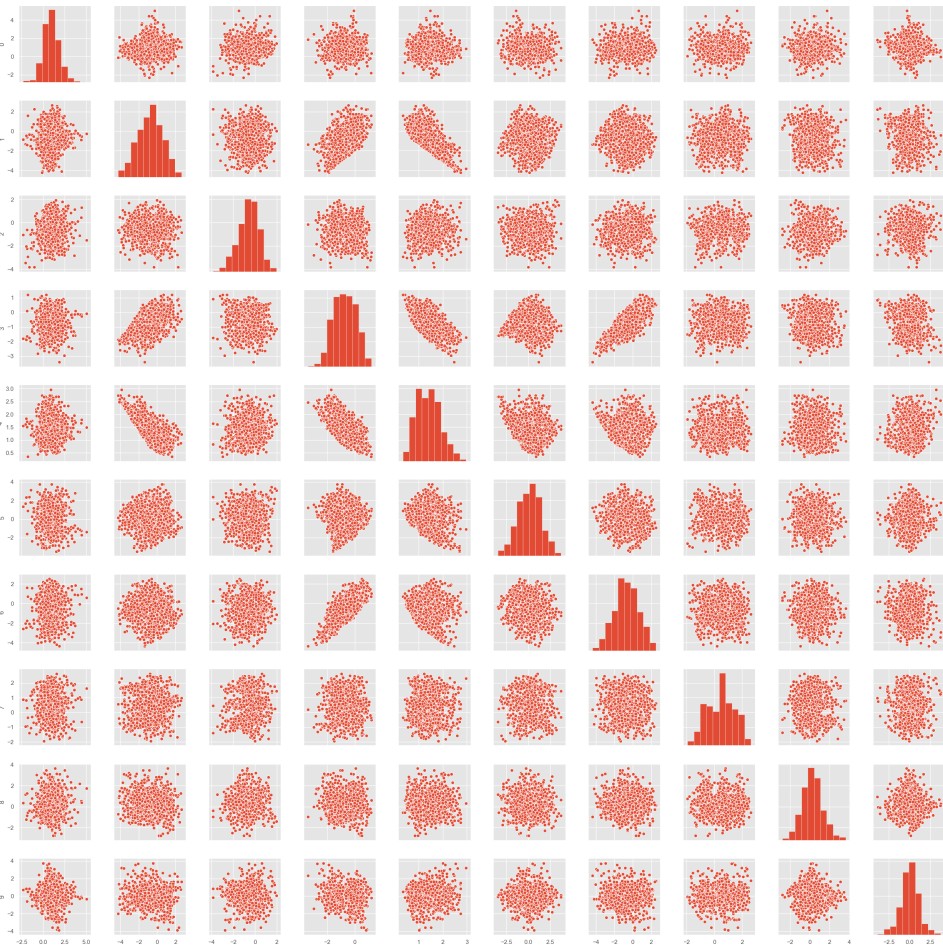

Figure 8: The pairplot of 10 latent dimensions of FactorVAE on Shapes3D. Some dimensions show strong correlation, e.g. dim 1&3, dim 1&4, dim 3&4, dim 3&6.

glement metric that considers *modularity, compactness* and *explicitness*. Then Ridgeway & Mozer (2018) made analysis on *compactness*, and *compactness* mean that each ground truth factor associates with only one or a few latent dimensions. They pointed out that in some situation a perfectly disentangled representation may not be *compact* (see Section. 3 in Ridgeway & Mozer (2018)).

Here we argue that *modularity* also should be reconsidered. A *modular* representation means that each dimension of latent space conveys information of at most one ground truth factor. This is exactly the goal attempted by Higgins et al. (2017); Kim & Mnih (2018); Chen et al. (2018), etc. However, multiple latent dimensions can work together to represent multiple ground truth factors meanwhile these latent dimensions are disentangled. For instance, $x$ and $y$ coordinates can be represented by $r$ and $\theta$ in polar coordinate system (or any coordinate system under rotation, i.e., $(x', y')^T = A(x, y)^T$ where $A$ is any orthogonal matrix). These coordinate systems are perfectly disentangled but $r$ (or $x'$) conveys information of both $x$ and $y$.

## 7 CONCLUSION

In this work, we demonstrated that our RTC-VAE, which rectifies the total correlation penalty can remedy its peculiar properties (disparity between total correlation of the samples and the mean representations). Our experiments show that our model has a more reasonable distribution of the mean representation compared with baseline models including $\beta$-TCVAE and FactorVAE. We also provide several theoretical proofs which could help diagnose several specific symptoms of entangle-

ment. Hopefully, our contributions could add to the explainability of the unsupervised learning of disentangled representations.

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

## A  APPENDIX

### A.1  MINIBATCH WEIGHTED SAMPLING (MWS)

See Chen et al. (2018),

$$\mathbb{E}_{q(z)}[\log q(z)] \approx \frac{1}{M} \sum_{i=1}^{M} \left[ \log \sum_{j=1}^{M} q(z(n_i)|n_j) - \log(NM) \right] \tag{11}$$

## A.2 MMS$_0$ AND MMS$_1$

There is no mathematical difference between MSS$_0$ and MSS$_1$ (same formulation as equation 5), only a difference in implementation. We replace this chunk of code ([https://github.com/rtqichen/beta-tcvae/blob/master/vae_quant.py#L199-L201](https://github.com/rtqichen/beta-tcvae/blob/master/vae_quant.py#L199-L201)) to

```
for i in range(batch_size):
        W[i,i] = 1/N
        W[i,(1+i)%batch_size] = strat_weight
```

## A.3 PROOF OF THEOREM 1

In the following proof, we use the convention of mathematical analysis that the meaning of C can change through lines to eliminate some redundant work of tracking.

**Theorem** (Theorem 1 restated). *Let $\boldsymbol{\mu} \sim \mathcal{N}(0, \boldsymbol{\Sigma})$ and $\sigma_j$ be the standard deviation of $\mu_j$, $j = 1, \cdots, D$, and $\max_j \sigma_j = c_0$. For a fixed $\mu$, let $\mathbf{z} \sim \mathcal{N}(\mu, \boldsymbol{\Sigma}'(\mu))$, where $\boldsymbol{\Sigma}'(\mu)$ is diagonal and satisfies that for some $R > 0$,*

$$
\begin{cases}
c_2 > \sigma'_j(\mu) > c_1 > 0, & \text{if } |\mu| < R, \\
c_3 > \sigma'_j(\mu) > \dfrac{c_4}{|\mu|^l}, \text{ for some } l \geq 1, & \text{if } |\mu| > R.
\end{cases}
\tag{12}
$$

*for some constants $c_1, c_2, c_3, c_4$. Then $\mathrm{TC}(\mathbf{z}) \leq C$ for some $C = C(R, c_0, \cdots, c_4, l) > 0$.*

Proof. Let

$$
S_+ = \{z \in \mathbb{R}^D | p(z) \geq \prod_j p(z_j)\}, \quad S_- = \{z \in \mathbb{R}^D | p(z) < \prod_j p(z_j)\},
$$

then

$$
\mathrm{TC}(\mathbf{z}) = \int_{S_+} + \int_{S_-} = \mathrm{TC}(\mathbf{z})_+ + \mathrm{TC}(\mathbf{z})_-.
$$

Since KL-divergence is non-negative, if $\mathrm{TC}(\mathbf{z})_+$ is bounded, then $\mathrm{TC}(\mathbf{z})$ must be bounded. In the following, we work on $S_+$, i.e., we assume $p(z) \geq \prod_j p(z_j)$.

For $|z| < R$,

$$
\begin{aligned}
p(z) &= \mathbb{E}_{p(\mu)}[p(z|\mu)] \\
&= \int p(\mu) \frac{1}{\sqrt{(2\pi)^D |\boldsymbol{\Sigma}'|}} e^{-\frac{1}{2}(z-\mu)(\boldsymbol{\Sigma}')^{-1}(z-\mu)^T} d\mu \\
&\leq \frac{1}{\sqrt{(2\pi)^D c_1^{2D}}} \int p(\mu) d\mu \\
&\leq \frac{C}{c_1^D},
\end{aligned}
$$

and

$$
\begin{aligned}
\prod_j p(z_j) &\geq \prod_j \int_{B_R(0)} p(\mu_j) \frac{1}{\sqrt{(2\pi)^D c_2^2}} e^{-\frac{(z_j - \mu_j)^2}{2c_1^2}} d\mu_j \\
&\geq \frac{C}{c_2^D} e^{-\frac{2R^2}{c_1^2}}.
\end{aligned}
$$

For $|z| > 2R$,

$$p(z) = \int_{B_R(0)} + \int_{B_{\frac{|z|}{2}}(0) \backslash B_R(0)} + \int_{B^c_{\frac{|z|}{2}}(0)}$$

$$\leq \frac{C}{c_1^D} e^{-\frac{||z|-R|^2}{2c_2^2}} + C \frac{|z|^{Dl}}{c_4^D} e^{-\frac{|z|^2}{8c_2^2}} + \int_{B^c_{\frac{|z|}{2}}(0)} p(\mu) \frac{e^{-\frac{1}{2}(\mu-z)(\mathbf{\Sigma}')^{-1}(\mu-z)^T}}{\sqrt{(2\pi)^d |\mathbf{\Sigma}'|}} d\mu$$

$$\leq \frac{C}{c_1^D} e^{-\frac{||z|-R|^2}{2c_2^2}} + C \frac{|z|^{Dl}}{c_4^D} e^{-\frac{|z|^2}{8c_2^2}} + \int_{B^c_{\frac{|z|}{2}}(0)} p(\mu) \frac{|\mu|^{Dl}}{\sqrt{(2\pi)^D} c_4^D} d\mu$$

$$\leq \frac{C}{c_1^D} e^{-\frac{||z|-R|^2}{2c_2^2}} + \frac{C}{c_4^D} |z|^{dl} e^{-\frac{|z|^2}{8c_2^2}} + \frac{C}{c_4^D} |z|^{Dl+D-2} e^{-\frac{|z|^2}{8c_0^2}}$$

$$\leq \frac{C}{r_1^D} |z|^{Dl+D-2} e^{-\frac{|z|^2}{8r_0^2}},$$

where $r_0 = \max(c_0, c_2)$ and $r_1 = \min(c_1, c_4)$, and since $l \geq 1$ and $D \geq 1$, $Dl + D - 2 \geq 0$. For $|z| \in (R_1, 2R_1)$, it is easy to see that $p(z) < C$. And for $|z| > R$,

$$\prod_j p(z_j) \geq \prod_j \left( \int_{B_{c_0}(0)} p(\mu_j) \frac{1}{\sqrt{(2\pi)^D c_2^2}} e^{-\frac{(z_j-\mu_j)^2}{2c_1^2}} D\mu_j \right)$$

$$\geq \frac{C}{c_2^D} e^{-\frac{2|z|^2}{c_1^2}}.$$

Hence,

$$\text{TC}(\mathbf{z}) = \mathbb{E}_{p(z)} \left[ \log \frac{p(z)}{\prod_j p(z_j)} \right]$$

$$\leq \int_{B_R(0)} p(z) \log C \frac{c_2^D}{c_1^d} e^{\frac{2R^2}{c_1^2}} dz + \int_{B^c_{2R}(0)} p(z) \log(C \frac{c_2^D}{r_1^D} |z|^{Dl+D-2} e^{\frac{2|z|^2}{c_1^2} - \frac{|z|^2}{2r_0^2}}) dz + C$$

$$\leq D \log \frac{c_2}{c_1} + \frac{2R^2}{c_1^2} + \int_{B^c_{2R}(0)} e^{-\frac{|z|^2}{8r_0^2}} [C + (Dl + D - 2) \log|z| + \frac{2|z|^2}{c_1^2} - \frac{|z|^2}{2r_0^2}] dz + C$$

$$\leq D \log \frac{c_2}{c_1} + \frac{2R^2}{c_1^2} + C.$$

$\square$

## A.4 PROOF OF PROPOSITION 1

**Proposition** (Proposition 1 restated)**.** *Let* $\mathbf{x} \sim \mathcal{N}(0, \mathbf{\Sigma})$*, then*

$$\text{TC}(\mathbf{x}) = \frac{1}{2} \left( \log|\text{diag}(\mathbf{\Sigma})| - \log|\mathbf{\Sigma}| \right). \tag{13}$$

Proof. First, recall that the KL-divergence between two distributions $\mathbb{P}$ and $\mathbb{Q}$ is defined as

$$\text{KL}(\mathbb{P}||\mathbb{Q}) = \mathbb{E}_\mathbb{P}[\log \frac{\mathbb{P}}{\mathbb{Q}}]$$

Also, the density function for a multivariate Gaussian distribution $\mathcal{N}(\mu, \mathbf{\Sigma})$ is

$$p(x) = \frac{1}{(2\pi)^{n/2} \det(\mathbf{\Sigma})^{1/2}} \exp(-\frac{1}{2}(x - \mu)^T \mathbf{\Sigma}^{-1}(x - \mu)).$$

Now, for two multivariate Gaussian $\mathbb{P}_1$ and $\mathbb{P}_2$, we have
$$\text{KL}(\mathbb{P}_1||\mathbb{P}_2) = \mathbb{E}_{\mathbb{P}_1}[\log\mathbb{P}_1 - \log\mathbb{P}_2]$$
$$= \frac{1}{2}\log\frac{\det\mathbf{\Sigma}_2}{\det\mathbf{\Sigma}_1} + \frac{1}{2}\mathbb{E}_{p_1(x)}[-(x-\mu_1)^T\mathbf{\Sigma}_1^{-1}(x-\mu_1) + (x-\mu_2)^T\mathbf{\Sigma}_2^{-1}(x-\mu_2)]$$
$$= \frac{1}{2}\log\frac{\det\mathbf{\Sigma}_2}{\det\mathbf{\Sigma}_1} + \frac{1}{2}\mathbb{E}_{p_1(x)}[-\text{tr}(\mathbf{\Sigma}_1^{-1}(x-\mu_1)(x-\mu_1)^T) + \text{tr}(\mathbf{\Sigma}_2^{-1}(x-\mu_2)(x-\mu_2)^T)]$$
$$= \frac{1}{2}\log\frac{\det\mathbf{\Sigma}_2}{\det\mathbf{\Sigma}_1} - \frac{1}{2}\text{tr}(\mathbf{\Sigma}_1^{-1}\mathbf{\Sigma}_1) + \frac{1}{2}\mathbb{E}_{p_1(x)}[\text{tr}(\mathbf{\Sigma}_2^{-1}((xx^T - 2x\mu_2^T + \mu_2\mu_2^T)))]$$
$$= \frac{1}{2}\log\frac{\det\mathbf{\Sigma}_2}{\det\mathbf{\Sigma}_1} - \frac{n}{2} + \frac{1}{2}\mathbb{E}_{p_1(x)}[\text{tr}(\mathbf{\Sigma}_2^{-1}((x-\mu_1+\mu_1)(x-\mu_1+\mu_1)^T - 2x\mu_2^T + \mu_2\mu_2^T))]$$
$$= \frac{1}{2}\log\frac{\det\mathbf{\Sigma}_2}{\det\mathbf{\Sigma}_1} - \frac{n}{2} + \frac{1}{2}\mathbb{E}_{p_1(x)}[\text{tr}(\mathbf{\Sigma}_2^{-1}((x-\mu_1)(x-\mu_1)^T + \underbrace{2(x-\mu_1)\mu_1}_{\mathbb{E}_{p_1(x)}(x)=\mu_1} + \mu_1\mu_1^T - 2x\mu_2^T + \mu_2\mu_2^T))]$$
$$= \frac{1}{2}\log\frac{\det\mathbf{\Sigma}_2}{\det\mathbf{\Sigma}_1} - \frac{1}{2}n + \frac{1}{2}\text{tr}(\mathbf{\Sigma}_2^{-1}(\mathbf{\Sigma}_1 + (\mu_2-\mu_1)(\mu_2-\mu_1)^T))$$
$$= \frac{1}{2}(\log\frac{\det\mathbf{\Sigma}_2}{\det\mathbf{\Sigma}_1} - n + \text{tr}(\mathbf{\Sigma}_2^{-1}\mathbf{\Sigma}_1) + (\mu_2-\mu_1)^T\mathbf{\Sigma}_2^{-1}(\mu_2-\mu_1))$$

Let $\mathbb{P}$ be a multivariate Gaussian $\mathcal{N}(\mu, \mathbf{\Sigma}_1)$, and then the product of the marginal distribution $\prod_i p_i(x)$ is also Gaussian $\mathcal{N}(\mu, \mathbf{\Sigma}_2)$, where $\mathbf{\Sigma}_2 = \text{diag}(\mathbf{\Sigma}_1)$. Thus, the total correlation of multivariate Gaussian distribution is
$$\text{TC}(\mathbf{x}) = D_{KL}(p(x)||\prod_i p_i(x))$$
$$= \frac{1}{2}(\log\frac{\det\mathbf{\Sigma}_2}{\det\mathbf{\Sigma}_1} - n + \text{tr}(\mathbf{\Sigma}_2^{-1}\mathbf{\Sigma}_1) + (\mu-\mu)^T\mathbf{\Sigma}_2^{-1}(\mu-\mu))$$
$$= \frac{1}{2}(\log\frac{\det\mathbf{\Sigma}_2}{\det\mathbf{\Sigma}_1} - n + n)$$
$$= \frac{1}{2}(\log|\text{diag}(\mathbf{\Sigma}_1)| - \log|\mathbf{\Sigma}_1|)$$

□

## A.5 PROOF OF EQUATION 10

Proof.
$$P(|z^{(i)} - \mu^{(j)}| < t) = P(|x| < t) \text{ where } \mathbf{x} \sim \mathcal{N}(0,2)$$
$$= \int_{-t}^{t}\frac{1}{\sqrt{4\pi}}e^{-\frac{x^2}{4}}dt$$
$$= \sqrt{\int_{-t}^{t}\frac{1}{\sqrt{4\pi}}e^{-\frac{x^2}{4}}dx\int_{-t}^{t}\frac{1}{\sqrt{4\pi}}e^{-\frac{y^2}{4}}dy}$$
$$= \sqrt{\int_{-t}^{t}\int_{-t}^{t}\frac{1}{4\pi}e^{-\frac{x^2+y^2}{4}}dxdy}$$
$$= \sqrt{\int_{0}^{2\pi}\int_{0}^{t}\frac{1}{4\pi}re^{-\frac{r^2}{4}}drd\theta}$$
$$= \sqrt{1 - e^{-\frac{t^2}{4}}}$$
$$= \frac{t}{2} + O(t^2)$$

□

## A.6 EXPERIMENTS

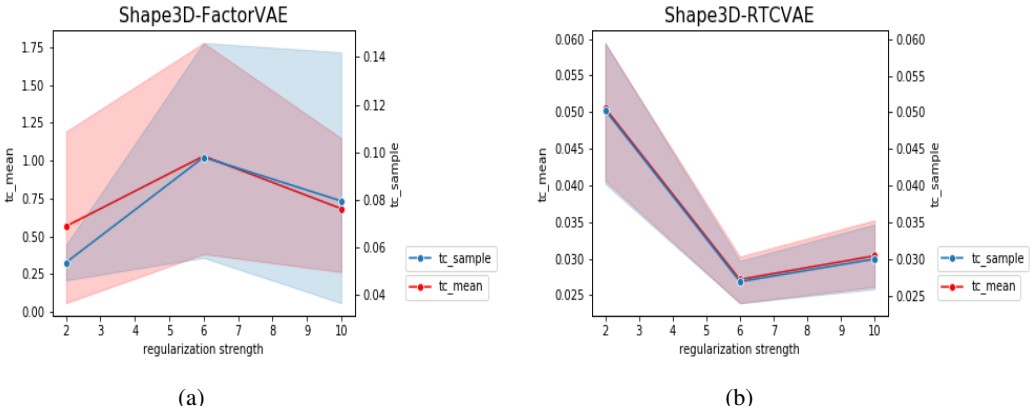

(a)                                   (b)

Figure 9: The shaded region indicates $90\%$ confidence interval. (a) The total correlation of sample and mean representation of FactorVAE on Shapes3D with $\beta = 2, 6, 10$. There is a large difference in scales of $TC_{sample}$ and $TC_{mean}$. (b) The total correlation of sample and mean representation of RTCVAE on Shapes3D with $\beta = 2, 6, 10$ and $\eta = 10$. There is almost no difference between $TC_{sample}$ and $TC_{mean}$ due to the variance penalty term in equation 4.

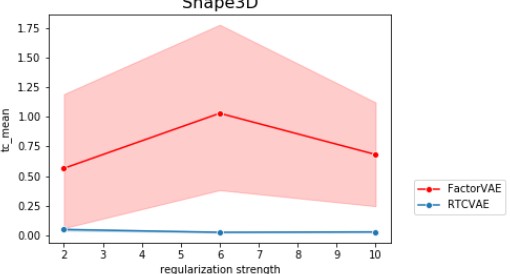

Figure 10: Direct comparison between the $TC_{mean}$ of FactorVAE and RTCVAE.

