# OpenReview forum: "RTC-VAE: HARNESSING THE PECULIARITY OF TOTAL CORRELATION  IN LEARNING DISENTANGLED REPRESENTATIONS"
_ICLR.cc/2020/Conference — Reject_

### Official Review · AnonReviewer3 · 2019-10-23
**Official Blind Review #3**

**Rating:** 3

**Review:**

This paper considers the extensions of variational autoencoders (VAEs), which take into account the total correlation of sampled distribution of latent variables. Proving a theorem that a family of distributions of sample representations with a bounded total correlation can have a mean representation of arbitrarily large total correlation, the authors propose RTC-VAE, which additionally penalizes total covariance of sampled latent variables. The authors demonstrate that RTC-VAE produces less correlated distributions of mean representation compared with baselines.

The proposed method, RTC-VAE, is based on a simple idea and its performance in experiments is promising. However, its derivation is somewhat ad-hoc and the experiments are not so comprehensive enough to provide the evidence for its good performance.

The covariance term in RTC-VAE of Eq. (4), is indirectly related to mean representation. Is it difficult to penalize TC(\mu), directly?

The arguments in Section 5 are somewhat superficial. It isn’t explained how Eq. (5) is derived. Without the explicit definition of D(z), Eq. (8) is not so informative.

Section 6: It is not discussed why the regularization parameter \eta is fixed to 10 and how RTC-VAE is sensitive to \eta.

Are there any other performance measures to indicate the good overall performance of RTC-VAE other than the training ELBO in Fig. 5?

Minor comments:
p.3, l.21 from the bottom: [z] should not be in the subscript.
p.4, l.3: n_{m+1} -> n_{M+1}
p.6, l.17 from the bottom: ``"that Factor VAE does" -> ``"than Factor VAE does"


The authors' responses answered my questions. However, I still think that the paper  has room for improvement in justifying the method, explaining the choice of hyperparameter and so on.


**Experience Assessment:**

I have read many papers in this area.

**Review Assessment: Checking Correctness Of Derivations And Theory:**

I assessed the sensibility of the derivations and theory.

**Review Assessment: Checking Correctness Of Experiments:**

I assessed the sensibility of the experiments.

**Review Assessment: Thoroughness In Paper Reading:**

I made a quick assessment of this paper.

---

> ### Author Response · Authors · 2019-11-15
> **Why do we add an additional regularizing term?**
>
> To penalize TC(mu) directly is exactly the work done by Chen et al.-2018 and Kim and Mnih-2018 with beta-TCVAE and FactorVAE. However, their approaches turn to have issues, pointed out by Locatello et al.-2019 The main point of our paper is to solve these issues of penalizing TC(mu) by adding an additional regularizing term.
>
> Equation (5) is derived in Chen et al. In equation (8), D is a discriminator as stated below equation (8).
>
> It is a good point that we should explain about our choice of hyperparameter beta and eta. The reason for eta=10 is simple. Since the variance term in equation 4 soon becomes small (close to 0) after training begins, this term will not contribute much compared with L_{beta-TC} term. So, eta=10 is enough to strengthen the penalty at the beginning of training. Higher value cannot bring further improvement since z is close to mu already. And lower values may not guarantee z being close to mu.
>
> It is a critical yet open question where there is an quantitative indicator of the performance of a VAE. Many people make various efforts in this area, however, there exist all sorts of issues for all of them, and none of them has been widely accepted as a good measurement, see Locatelo et al-2019. As a result, we measure our models by total correlation, ELBO and qualitative analysis.
>
> All minors are fixed.

---

### Official Review · AnonReviewer2 · 2019-10-23
**Official Blind Review #2**

**Rating:** 1

**Review:**

The paper proposes a method to address a known problem for unsupervised disentangling methods that penalises total correlation, namely that while the total correlation of the samples from q(z) (denoted TC(z)) are encouraged to be small, the total correlation of the means of q(z|x) (denoted TC(mu)), used as the disentangled representation in practice, is not necessarily small and can increase with regularisation strength.

In the introduction, I think that the statement “they concluded pessimistically that it is fundamentally impossible to learn a disentangled representation in an unsupervised setting” is a wrong interpretation of Locatello et al. They show that optimising marginal likelihood in a generative model (such as a VAE) cannot achieve disentangling without any inductive biases in the model. But there inductive biases in the models used by disentangling methods, along with the loss (that is a variant of the ELBO and not the marginal likelihood), that allow disentangling in practice. There are also theoretical works such as [1] that explain this behaviour.

The theoretical contribution of the paper is Theorem 1, that claims to show the existence of distributions with arbitrarily large TC(mu) but with small, bounded TC(z). Indeed the proof shows that TC(z) is bounded by C, but looking at Appendix A it seems as though C is a function of c1,c2 and R that is used to define p(z|mu), and is lower bounded by (2pi)^{-D/2} (a constant, which confusingly, is also denoted by C in the appendix). It appears necessary to have another line that mentions how small C can be chosen to be via choice of c1,c2,R. Also it seems as though the proof can be largely simplified by having sigma’_j(mu)=c_1 if |mu|<R and sigma’_j(mu)=c_4/|mu| if |mu|>R, removing free parameters l,c_2,c_3.

The methodological contribution of the paper is to propose an extra regularisation term that penalises the variances of q(z|x). While this does introduce another hyperparameter to tune, it has the advantage of being simple to implement and having an intuitive explanation of how it can address the problem; if q(z|x) is encouraged to have smaller variance, the distribution of z will be encouraged to be closer to the distribution of mu, hence helping to address the disparity between TC(z) TC(mu).

I like the simplicity of the idea, however the analysis is lacking in rigour. First of all, when comparing the different methods of estimating TC(z), it’s not clear what the mathematical difference of MSS_0 and MSS_1 is. This should be explicitly stated so that one can understand the results in Figure 1. Regarding the following analysis, it’s not clear why the off diagonal elements of the cube (i.e. q(z^(i)_k|n^(j)) when i != j ) should be very small compared to the diagonal elements. For example, it could be the case that z^(i) and z^(j) are close to one another, in which case the (i,j)th entries will be non-negligible compared to the diagonal. Hence the analysis is difficult to accept. Also the claim that MSS and MWS prefer to shut down latent dims should be verified by experiments. Further, it’s unclear why this is undesirable from a disentangling point of view. Of course we don’t want to use fewer number of latent dimensions than the number of ground truth factors, but also we don’t want to use more latent dimensions. Also the at the bottom of page 5 is a Gaussian with a correlated covariance matrix, and it’s claimed that its TC can be arbitrarily large, but surely this is fixed? Finally the authors list lots of reasons why not to use MSS, but there is no discussion about the density-ratio trick method for estimating TC. For example, it is known that this method suffers underestimates of TC (c.f. Kim & Mnih) - it has its own issues, that can be arguably more severe than MSS. This analysis needs a lot more explanation and rigour.

The experimental results are very weak and sparse, that is nowhere near enough to give a convincing case for the newly proposed method. The method has only been trained on dsprites and 3d shapes, with three choices of beta and a single value of eta, and only estimates of TC(mu) and TC(z) are reported, with no evaluation of disentanglement performance. There is some evidence in the paper that the regularisation makes both TC(mu) and TC(z) close to 0 with eta=10, but it’s unclear how this affects disentangling performance, and whether smaller values of eta can give a sweetspot. The experiments should cover a larger range of datasets, with evaluation on how different disentanglement metrics, TC(mu) and TC(z) change for different values of beta and eta, along with a comparison with other disentangling methods, especially DIP-VAE-1, that directly penalises correlation in mu (for an open source library that facilitates this, see e.g. github.com/google-research/disentanglement_lib). Even the authors acknowledge that “the scale of our experiments is limited”, and it is clear that the paper is not yet ready for publication.

Overall, the proposed idea is simple and easy to implement, which is the main advantage of the paper, but it is evident that the analysis and evaluation lacks rigour, hence the paper will need to undergo significant revision to be in a publishable state.

[1] Rolinek, M., Zietlow, D. and Martius, G., Variational Autoencoders Pursue PCA Directions (by Accident). CVPR 2019.

Minor typos/comments:
Eqn(4): the ||.||_1 should be replaced by trace, since the term inside ||.||_1 is a covariance matrix (although it’s diagonal). If A is a matrix, ||A||_1 is the maximum absolute column sum, which is different to what is meant by the paper, the trace (sum of diagonals).
p3: Can MWS also be stated in the paper (or at least in the appendix) to make it self-contained?
p4: followed <- follow, n_{m+1} <- n_{M+1}
p6-7: Section 6.1 should be in a related work section, and not under the Experiment section. The point about modularity is a fair but known issue, closely related to the issue of axis alignment/unidentifiability (see e.g. Rolinek et al)


**Experience Assessment:**

I have published one or two papers in this area.

**Review Assessment: Checking Correctness Of Derivations And Theory:**

I assessed the sensibility of the derivations and theory.

**Review Assessment: Checking Correctness Of Experiments:**

I carefully checked the experiments.

**Review Assessment: Thoroughness In Paper Reading:**

I read the paper thoroughly.

---

> ### Author Response · Authors · 2019-11-15
> **A few respectful responses**
>
> Dear reviewer, thank you for your profound review and all the valuable insights.
>
> We agree with your understanding of Locatello et al.’s result and revised the related sentence. All the VAEs they considered are based on optimizing marginal likelihood, which means their conclusion should be limited to this scope. Note that RTCVAE proposed by us is still one of this kind, and our objective does not introduce inductive biases. It is one of our main points that some issue raised by them can be solved in our paper, hence to challenge their Challenging Common Assumptions… Thank you for introducing the work about inductive biases done by Ronelik et al., which is very inspiring and helpful.
>
> On your comments on Theorem 1:
> 1.	Regarding the simplification of the proof, actually we started our proof with these parameters, R, c_0,… c_4, l, and did not intentionally try to expand the generality of the theorem. Instead, we were surprised to see that there exist such a large class of distributions of z that all have bounded TC(z). When the objective function only penalizes TC(mu), neural networks are so flexible to easily find a distribution with low TC(z), and total correlation estimators like MSS can encourage shutting down latent dimensions, which together cause the disparity of TC(mu) and TC(z). This fact was not noticed until Locattelo et al.’s work, and our investigation gives an explanation of the cause. Hence, Theorem 1 leads to the necessity of penalizing the difference between the distributions of mu and z, e.g. equation (4), simple as it is, yet necessary in our opinion.
> 2.	Can the bound C of TC(z) be arbitrarily close to 0 by tweaking parameters R, c_0,… c_4, l? No. It is an interesting question whether there exists a distribution of z~ N(mu, Sigma’(mu)) with arbitrarily small TC(z) given mu. However, this question remains open to us for now.
> 3.	Sorry we don’t understand what you mean by “C…is lower bounded by (2pi)^{-D/2}”. Does TC(z) have a lower bound greater than 0? This will be the next question, if the answer to the previous question is no. In other words, if TC(z) cannot be arbitrarily small, what’s the lower bound? We leave it to future work.
> 4.	As for the notation of C, it is a convention that throughout the mathematical analysis, the meaning of C can change to eliminate some redundant work of tracking. For example, in the last two lines of our proof of Theorem 1, C absolves the integral in the front (the 3rd term in the second last line) since the integral is also bounded. We apologize for the confusion and will make some changes to reduce that.
>
> Your concern about our argument of MSS_0 and MSS_1 is valuable to us, and we will revise this part to make it clearer for readers.
>
> 1.	There is no mathematical difference between MSS_0 and MSS_1 (same formulation), only a difference in implementation. See Appendix A.2
> 2.	For the part that is difficult for you to accept, we add more explanation, see equation (10). In short, we show that the chance of z^(i) and z^(j) (actually it should be mu^(j)) to be close is very small in high dimensional latent space.
> 3.	We have observed from experiments that MSS/MWS shut down dimensions, and will list some results in our paper. Shutting down dimension is undesirable firstly due to the reason you mentioned, i.e., latent dimension should not be fewer than ground truth. Moreover, it is very natural for VAE to learn more latent dimensions than ground truth. Take dSprites for example, shape may need more than one dimension to describe as ground truth does. We will address this further in our revision.
> 4.	On the bottom of Page 5, there is a type, a “)” is missing after the correlation matrix Sigma. We just gave an example of strongly correlated z in the parentheses. We revised this part to make it clearer.
> 5.	We observed much more stable estimation of TC with the density trick than with MSS/MWS. Though the former is known to have issues, we don’t have further insights to share. Meanwhile, we pointed our some previously unknown issues of the latter that people should consider before using these techniques.
>
> In addition to dSprites and 3d shapes, we also trained our model on 3d faces. We agree with your opinion that we will included more experiments to this section.
> 1.	Choosing eta=10 is because the variance of z|mu actually decays very fast and then stops around 0. Since our purpose is just to eliminate the disparity of z and mu, eta=10 serves this purpose. Higher value of eta cannot contribute more to the loss function since variance is close to 0, while lower value is insufficient.
> 2.	We take your suggestion on DIP-VAE-I seriously and plan to do it later. Unfortunately we didn’t have the time to do the comparison before submission.
>
> All minors are fixed.

---

### Official Review · AnonReviewer1 · 2019-10-23
**Official Blind Review #1**

**Rating:** 3

**Review:**


The novelty of this paper is adding an extra regularization term to the objective of beta-TCVAE (a VAE that regularizes total correlation), based on the discovery that low TC(z) does not necessarily mean low TC(mu). The added term enforces sample and mean representations stay close.

The authors' idea is understandable at a coarse resolution. However, the authors explain the mathematics poorly. Explanations of lots of variables and notions are missing. For example in Theorem 1, what is "j"? what is \sigma_j? In Section 4, the simplification of notations lead to more difficulties to understand the formulas. In "x_n", is n the index of a sample or a dimension? The notations of variables are also confusing. Boldface lowercase letters should be used for vectors, and plain letter should be used for scalars. In Equation 4, what are D and k?

It is nice to see, in the given experimental results,  that latent representations of RTCVAE are less correlated in comparison with FactorVAE in Figures 6 and 7. However, the authors should show some generated examples through latent variable traversal to qualitatively demonstrate the potential advantages of the proposed improvement.

Minors: Section. X -> Section X



**Experience Assessment:**

I have published one or two papers in this area.

**Review Assessment: Checking Correctness Of Derivations And Theory:**

I assessed the sensibility of the derivations and theory.

**Review Assessment: Checking Correctness Of Experiments:**

I carefully checked the experiments.

**Review Assessment: Thoroughness In Paper Reading:**

I read the paper at least twice and used my best judgement in assessing the paper.

---

> ### Author Response · Authors · 2019-11-15
> **About math notations**
>
> Thank you for your feedback. sigma_j is the std of each mu_j. Hence j in the index of latent dimensions. x_n is a sample, and n is indeed the index of sample. Bold letters here indicate both a vector and a random variable. For example, mu is a vector in latent space, and mu is also a random variable encoded from a sample x. In equation (4), D is the dimension of latent space and k is the index of the dimension.
>
> We will provide traversal examples, and they were missing due to a very rush submitting.
>
> All minors are fixed.

---

### Decision · Program_Chairs · 2019-12-19

**Decision:**

Reject

**Comment:**

This paper highlights the problem of penalizing the total correlation of sampled latent variables for unsupervised learning of disentangled representations. Authors prove a theorem on how sample representations with bounded total correlation may have arbitrarily large total correlation when computed with the underlying mean. As a fix, the authors propose RTC-VAE method that penalizes total covariance of sampled latent variables.

R2 appreciated the simplicity of the idea, making it easy to understand and implement, but raises serious concerns on empirical evaluation of the method. Specifically, very limited datasets (initially dsprites and 3d shapes) and with no evaluation of disentanglement performance and no comparison against other disentangling methods like DIP-VAE-1. While the authors added another dataset (3d face) in their revised versions, the concerns about disentanglement performance evaluation and its comparison against baselines remained as before, and R2 was not convinced to raise the initial score.

Similarly, while R1 and R3 appreciate author's response, they believe the response was not convincing enough for them, and maintained their initial ratings.

Overall, the submission has room for improvement toward a clear evaluation of the proposed method against related baselines.